# Effect of Intensive Rehabilitation Program in Thermal Water on a Group of People with Parkinson’s Disease: A Retrospective Longitudinal Study

**DOI:** 10.3390/healthcare10020368

**Published:** 2022-02-14

**Authors:** Roberto Di Marco, Francesca Pistonesi, Valeria Cianci, Roberta Biundo, Luca Weis, Lucrezia Tognolo, Alfonc Baba, Maria Rubega, Giovanni Gentile, Chiara Tedesco, Miryam Carecchio, Angelo Antonini, Stefano Masiero

**Affiliations:** 1Parkinson and Movement Disorders Unit, Study Centre on Neurodegeneration (CESNE), Department of Neuroscience, University of Padova, Via Giustiniani 5, 35128 Padova, Italy; francesca.pistonesi@unipd.it (F.P.); valeria.cianci@unipd.it (V.C.); luca.weis@unipd.it (L.W.); giovanni.gentile@unipd.it (G.G.); chiara.tedesco@unipd.it (C.T.); miryam.carecchio@unipd.it (M.C.); angelo.antonini@unipd.it (A.A.); 2Department of Neuroscience, School of Physical Medicine and Rehabilitation, University of Padova, Via Giustiniani 2, 35128 Padova, Italy; lucrezia.tognolo@unipd.it (L.T.); maria.rubega@unipd.it (M.R.); stef.masiero@unipd.it (S.M.); 3Physical Medicine and Rehabilitation Unit, University Hospital of Padova, Via Giustiniani 2, 35128 Padova, Italy; alfonc.baba@aopd.veneto.it; 4Department of General Psychology, University of Padova, Via Venezia 8, 35131 Padova, Italy; roberta.biundo@unipd.it

**Keywords:** Parkinson’s disease, physical therapy, aquatic therapy, motion analysis

## Abstract

The main objective of this study is to test the effect of thermal aquatic exercise on motor symptoms and quality of life in people with Parkinson’s Disease (PD). Fourteen participants with diagnosis of idiopathic PD completed the whole rehabilitation session and evaluation protocol (Hoehn and Yahr in OFF state: 2–3; Mini Mental State Examination >24; stable pharmacological treatment in the 3 months prior participating in the study). Cognitive and motor status, functional abilities and quality of life were assessed at baseline and after an intensive rehabilitation program in thermal water (12 sessions of 45 min in a 1.4 m depth pool at 32–36 ${}^{\circ }$C). The Mini Balance Evaluation System Test (Mini-BESTest) and the PD Quality of Life Questionnaire (PDQ-39) were considered as main outcomes. Secondary assessment measures evaluated motor symptoms and quality of life and psychological well-being. Participants kept good cognitive and functional status after treatment. Balance of all the participants significantly improved (Mini-BESTest: p<0.01). The PDQ-39 significantly improved after rehabilitation (p=0.038), with significance being driven by dimensions strongly related to motor status. Thermal aquatic exercise may represent a promising rehabilitation tool to prevent the impact of motor symptoms on daily-life activities of people with PD. PDQ-39 improvement foreshows good effects of the intervention on quality of life and psychological well-being.

## 1. Introduction

Parkinson’s Disease (PD) is the second most common neurodegenerative disease and is characterised by cognitive deterioration [1,2] and impaired gait and postural control [3,4,5,6,7,8], leading to an increased risk of falls [9]. Pharmacological treatment, complemented with physical exercise, hampers the impact of motor symptoms in daily life of people with PD [10]. However, effectiveness of physical therapy on social and psycho-social aspects, and quality of life is still debated [11].

Despite its known benefits, land-based exercise may call for fast turns or stepping on uneven surfaces that may trigger the Freezing of Gait in individuals with PD, inducing possible unsafe situations [12]. Aquatic-based exercise has the intrinsic advantage of helping individuals lightening their body weight and reducing the risk of harm and falls during the exercises [13,14], potentially leading to a higher effectiveness of the exercise itself [15]. Aquatic exercise has been used in older adults with impaired physical functions as well as neuromuscular and orthopaedic diseases affecting balance and mobility [14,16,17]. In people with PD, water viscosity is an additional sensory cue that facilitates therapy effectiveness on dynamic balance [18,19,20,21,22].

The Euganean basin of Abano and Montegrotto (Italy) is rich of hyperthermal salso-bromo-iodic water, which is markedly hypertonic, has anti-inflammatory properties and promotes the release of endorphins [23]. The WHO defined the thermal aquatic exercise as traditional and complementary medicine [24]. Thermal aquatic exercise combines the benefits of the physical effect of immersion with the chemical effect of minerals [18,23,25]. The intrinsic high temperature of thermal water (32–36 ${}^{\circ }$C) is effective in pain management [26] and promotes the release of muscle tension [23], typical secondary symptoms that impact on psycho-social aspects and quality of life of people with PD. The thermal environment is also non-competitive and relaxing, leading to possible benefits on individuals’ quality of life and psychological well-being [27].

The aim of this research was to investigate the effect of thermal aquatic exercise on motor symptoms and quality of life and psychological well-being in people with PD. The main effect is expected to be observed on the scores evaluating motor symptoms. A reduced impact of PD on participants’ quality of life is also expected as an effect of the management of pain, the relase of muscle tension and the relaxing and non-competitive environment of thermal pools.

## 2. Materials and Methods

### 2.1. Participants and Ethics Statement

Twenty participants have been recruited at the Parkinson and Movement Disorders Unit of the University Hospital of Padova from September 2019 to December 2019. Participants that met the following inclusion criteria were included: (i) diagnosis of idiopathic PD according to the UK Parkinson’s Disease Society Brain Bank diagnostic criteria [28]; (ii) Hoehn and Yahr stage II-III in OFF-state [29]; (iii) Mini Mental State Examination (MMSE) score > 24 [30]; (iv) stable pharmacological treatment for the 3 months prior participating in the study and during the rehabilitation period.

Exclusion criteria were selected to avoid including confusion factors in the analysis, such as alterations in motor abilities not associated with PD. Participants were excluded if they suffered from/had: (i) deep brain stimulation and infusion therapies; (ii) diabetes; (iii) history of stroke; (iv) reported pathologies of the musculoskeletal system; (v) uncontrolled blood hypertension; (vi) severe cardiac and/or lung diseases—e.g., cancer; (vii) history of epileptic seizures; (viii) depression or severe psychiatric disorders; (ix) reported urinary incontinence; and (x) relevant brain comorbidities or cerebrovascular disease, as assessed with clinical T1w3D and FLAIR MRI protocol. Considering that participants were recruited and underwent the rehabilitation during the COVID-19 pandemic, a further exclusion criteria was the ascertained or the possible positivity to SARS-CoV-2 that may have neurological implications [31,32,33,34]. A triage questionnaire was administered to the participants before each visit. It was checked whether: (i) their body temperature was lower than 37.5 ${}^{\circ }$C; (ii) they experienced COVID-19 symptoms; (iii) their family members or close contacts (less than 1 m in the same room for at least 15 min without protective equipment) confirmed or suspected to have contracted COVID-19 in the 20 days prior the evaluation; (iv) they have ever been tested for SARS-CoV-2, especially in the 20 days prior the evaluation; and eventually if (v) they have ever been positive to SARS-CoV-2 tests. None of the patients included in this study had history of COVID-19 or had contact with positive people.

Two evaluation visits were carried out immediately before (baseline visit) and after (follow-up 1 visit) the 6-week rehabilitation session (Figure 1). Each visit aimed to evaluate participants’ anthropometry, functional abilities, quality of life and motor symptoms and was performed at the same day time during participants ON-phase (e.g., participant 001 was assessed at 9.00 am at both the baseline and follow-up visits).

The study was carried out according to the Declaration of Helsinki and ethical approval to data collection was granted by the local review board (AO/75048/3072). Participants read and signed an informed consent before participating in the study.

### 2.2. Rehabilitation Program

The intervention consisted of 12 sessions lasting 45 min (twice a week for 6 weeks) and conducted in a 1.4 m depth pool in a thermal rehabilitation centre. Participants were arranged in groups of five, with two physical therapists supervising each session. The intervention was always delivered to participants during their ON-medication state. The main objective of the thermal aquatic program was focused on improving balance, posture and gait (Table 1). With water viscosity serving as additional proprioceptive feedback, exercises promoted participants’ muscle strengthening and their consciousness on: (i) how to control their standing posture and trunk attitude; (ii) how to control intentional tremors; (iii) how to respond to external perturbation; (iv) how much power needs to be generated to shift their body weight forward and to plan the step length; (v) improving coordination between arms and legs opponent movements while standing and walking; and (vi) step-to-step consistency.

### 2.3. Clinical Assessment

Participants’ clinical description (age at disease onset, disease duration, age, gender, anthropometry) and Hoehn and Yahr (H&Y) Stage [29]) were collected by a neurologist experienced in the field of movement disorders. Levodopa (LEDD) and dopamine agonist equivalent daily (DAED) doses were calculated [35], and the presence of ongoing anticholinergic treatments was recorded.

### 2.4. Cognition, Functional Abilities and Quality of Life

Two experienced neuropsychologists administered all the following questionnaires and tests. To ensure consistency of the results, the same neuropsychologist interviewed each participant at baseline and follow-up visits. The MMSE was administered at the screening only, to ensure participants met the inclusion criteria (MMSE > 24).

The Montreal Cognitive Assessment (MoCA) was administered to evaluate the global cognitive status of the participants, investigating also attention and executive function (i.e., frontal-striatal function) and the visual spatial abilities, which are known to be altered even in early stage PDs [36,37]. In order to avoid learning effects, alternative MoCA versions were adopted at baseline and follow-up visits.

As further screening for dementia, the functional status of the participants was assessed via the Activities of Daily Living (ADL) [38] and the Instrumental Activities of Daily Living (IADL) [39] questionnaires. Subjective cognitive complaints and the impact of cognitive impairment without dementia on daily functioning were assessed using the Parkinson’s Disease Cognitive Functional Rating Scale (PD-CFRS) [40]. PD-CFRS is a 5-min questionnaire that explores a wide range of functional aspects associated with instrumental activities of daily living suspected to be sensible to cognitive impairment in PD, but not testing for the impact of motor symptoms on those activities [40].

Finally, quality of life was scored via the Parkinson’s disease Quality of Life Questionnaire (PDQ-39) [41]. The PDQ-39 consists of 39 questions grouped into 8 dimensions: i.e., Mobility, Activity of daily living (PDQ-39 ADL), Emotional well-being, Stigma, Social Support, Cognition, Communication and Bodily discomfort. It is worthy to underline that the PDQ-39 ADL differ from ADL evaluated via the PD-CFRS. Indeed, the PDQ-39 ADL quantifies the impact of motor symptoms on individuals’ activities of daily living [41].

### 2.5. Motor Assessment

After having completed the New Freezing of Gait Questionnaire (NFOG-Q) [42] to self report presence and severity of freezing of gait episodes (if any) in their daily lives, a neurologist collected the Unified Parkinson’s Disease Rating Scale part III (MDS-UPDRS part III) [43].

Participants’ static and dynamic balance control was evaluated via the Mini Balance Evaluation System Test (Mini-BESTest) [44,45,46], also used to predict risk of falls [47]. The Mini-BESTest includes four sections: anticipatory postural adjustments (APA), reactive postural control (RPC), sensory orientation (SO) and dynamic gait (DG). The SO section includes stabilometry tests on different surfaces (namely hard, soft and 30${}^{\circ }$ inclined) with both eyes-open (EO) and eyes-closed (EC) conditions and exploring both single and dual-task paradigms (DT). The DG section, instead, includes of Timed Up and Go (TUG) with and without dual-tasking to determine the effects of cognitive load on gait performance [48,49,50,51]. In the first three sections of the test, the maximum score is 6 and for the latter is 10, with a maximum total score of 28 points [47]. As part of the Mini-BESTest, participants were asked to perform the Time Up and Go test as measure of fall risk and measure of improvement of balance, sit to stand and walking. TUG test was also performed carrying a glass full of water without pouring it (i.e., TUG-ManualDT) to test gait performances during two motor tasks. The TUG test is a predictor of risk of falls in PD, with one being addressed as potential faller when the total score is TUG time is equal or larger than 11.5 s [52].

Lastly, participants were instructed to “cover the maximum possible distance in six minutes of walking” back and forth on a 10 m obstacle free walkway (six minutes walk distance, 6MWD) [53], “turning around the cones placed at the beginning and end of it without stopping”. The six minutes walk test (6MWT) was used to assess participants’ aerobic capacity and endurance [53].

### 2.6. Statistical Analysis

Having considered the Mini-BESTest score as primary outcome of this study, an a priori sample size estimation was conducted using G*Power v3.1.9.4 [54]. We considered, as reference, the effect size ($d_{z}=1.127$) obtained on Mini-BESTest score after standard rehabilitation in patients with Parkinson’s disease in [55]. An alpha probability error of 0.05 was considered for sample estimation. Results showed that a total sample of 13 participants with Parkinson’s disease was required to achieve a power of 0.95 in Mini-BESTest score using a two-tailed paired Wilcoxon signed-rank test. In order to take into account possible dropout a larger sample was recruited.

Statistics were calculated using RStudio (version 1.4.1106, Boston, MA, USA). For each variable, Minimal Detectable Changes (MDC) derived from the literature was considered to evaluate the noticeable change in ability [56]: Mini-BESTest (MDC = 4.1 [57]); MDS-UPDRS part III (MDC = 4.63 [58]), 6MWD (MDC = 82 m [59]); TUG (MDC = 3.5 s [60]).

Variables were tested for normal distribution via the Lilliefors test [61]. Pre-post rehabilitation effects were assessed with two-tailed paired Wilcoxon signed-rank test for variables that did not satisfied data normality, or the two-tailed paired t-test when data normality was verified. Statistical significance was set at p<0.05 and a Bonferroni correction was used for multiplicity adjustment. Effect size power analysis was calculated using Cohen’s $d_{z}$ measure correcting for small sample size and between-repetitions Pearson correlation [62].

For the Mini-BESTest and the PDQ-39, statistical differences were tested also among subitems. Distribution of changes observed between visits on each variable were described using the forest plots.

## 3. Results

### 3.1. Demographic and Clinical Data

Among those recruited, eighteen participants met the inclusion criteria and were included in the study. Three participants were evaluated at the baseline but did not feel confident to take part in the study due to the ongoing COVID-19 pandemic and did not start the rehabilitation program. One participant did not complete the rehabilitation program due to an ankle sprain not associated with the treatment itself and was not evaluated post-rehabilitation. None of the participants reported any relevant adverse event associated with the rehabilitation program. Table 2 shows the social and demographic characteristics of the fourteen participants that were included and completed the whole evaluation and rehabilitation protocol (70% of recruited participants). Among these only two participants had difficulties adhering to the physical therapist indications during rehab sessions: the 80% of those included after screening were thus completely compliant.

### 3.2. Cognition, Functional Abilities and Quality of Life

The functional status as assessed by ADL, IADL and PD-CFRS was preserved and remained similar at baseline and after rehabilitation (Table 3 and Figure 2a). Similar results were obtained for the MoCA corrected score.

The PDQ-39 score detected significant changes after therapy in the participants’ quality of life (p=0.029; Cohen’s $d_{z}=0.668$; mean difference equal to 5.36; 95% confidence interval: −9.99 to −0.73). This result is driven by the Mobility and ADL (*p*-values equal to 0.040 and 0.049, respectively). See Figure 2b and Figure 3a.

### 3.3. Motion Data

No significance was detected looking at NFOG-Q scores pre- and post-rehabilitation, with a null median difference among participants and between visits Figure 2a. Differences obtained for other motor scales and tests are reported in Table 3 and highlighted in Figure 3b, c, d and Figure 2c. In particular, the MDS-UPDRS part III score improved with a mean change equal to −8.77 (95% confidence interval: −15.78 to −1.77) from baseline to follow-up visit (p=0.022), with six participants mainly causing such significance (Figure 3d and Figure 2c).

The Mini-BESTest total score significantly improved, even after the Bonferroni correction, with a mean change equal to 3.79 (95% confidence interval: 2.18 to 5.39; p=0.0002). This result was mainly driven by the Reactive Postural Control (even though not significant, p=0.060) and Gait subitems (p=0.0008). See Figure 2c and Figure 3b.

The TUG test (Figure 2c and Figure 3c) highlighted significant differences in the time to complete the task in both single and dual-task conditions ($p_{TUG}=0.002$ and $p_{TUG-DT}<0.001$), even with the Bonferroni correction, but not when asking the participants to perform two motor tasks contemporary (TUG-ManualDT). TUG duration mean change was equal to −3.5 s (95% confidence interval: −9.28 to 2.25 s). Similar improvement was obtained for the TUG-DT, with a mean change equal to −3.87 s (95% confidence interval: −6.17 to −1.57 s). The outlier values (Figure 3c) are obtained from a participant who experienced a prolonged freezing of gait in all TUG tests during both visits.

Similarly, the 6MWT confirmed an improvement in participants’ aerobic capacity and endurance (Figure 2c and Figure 3d). The 6MWD augmented by 51 m (95% confidence interval: 8 to 94 m; p=0.025).

### 3.4. Response Rate for Balance and Gait

Based on the minimal detectable change for MDS-UPDRS part III (MDC = 4.63 [58]), an overall improvement in motor status was observed in 6/14 participants. The rehabilitation improved the Mini-BESTest score in 7 out of 14 participants considering a MDC equal to 4.1 [57], although the total score increase after rehabilitation in all patients (Figure 3b). Five participants also improved in the 6MWD after the rehabilitative program in thermal water, having considered a MDC equal to 82 m [59]. TUG timing was significantly reduced comparing baseline and follow-up 1 evaluation (p=0.002). Five participants performed the TUG at baseline with a timing over the threshold for being considered at risk of falling (11.5 s). After the rehabilitation, all of them improved the TUG timing, and two of those crossed the threshold for not being considered at risk of falling anymore.

## 4. Discussion

In this study, we aimed at investigating the effect of thermal aquatic exercise on motor symptoms and quality of life and psychological well-being in PD. As opposed to previous publication [18], we controlled the time of intake of medications of the participants both during the whole period of assessment (i.e., from baseline to follow-up) and during the intervention (i.e., aquatic exercise was delivered during their ON medication state).

The intervention (twelve sessions of 45 min) did not have an impact on cognitive performance (MoCA results, Figure 2a). Indeed, scores of ADL, IADL and PD-CFRS questionnaires at baseline are supportive of almost preserved cognitive status in all participants and therefore rehabilitation is not expected to further improve already maintained functional independence [11]. However, the thermal aquatic exercise seemed to positively impact on self-reported functional abilities as captured by the PDQ-39 ADL subitem and, thus, on the participants’ perceived quality of life (Table 3).

Balance of all the participants significantly improved as assessed via the Mini-BESTest score (Figure 2c and Figure 3b) with a large effect size (Cohen’s $d_{z}$ equal to 1.362). This result was mainly driven by the Gait section of the Mini-BESTest (Cohen’s $d_{z}$ equal to 1.167), which aims to evaluate adaptability of walking to perturbations. Improvements in this section reflect improvements in dynamic balance control with particular attention to gait (Figure 2c and Figure 3c). Although not significant, results obtained for the Mini-BESTest reactive postural control section are in line with this finding, showing a trend of improvement in the ability to answer to external balance perturbations in static conditions. The MDS-UPDRS part III score and the six minutes walk distance also reflected the improvement in the participants’ motor status. A Mini-BESTest total score lower than 17.5 has been previously used as predictor of high fall risk [63]. Seven of the included participants were below this cut off before the intervention, whereas only one participant was classified as potential faller at the post-intervention visit (total score at baseline equal to 8—the worst score—and 11 at the follow-up). Our results thus confirm that thermal aquatic exercise hampers the impact of motor symptoms in daily life activities of people with PD [10], also reducing risk of falls [64,65,66,67], most likely thanks to body weight lightening [13,14], viscosity as additional sensory cue [18] and the high temperature promoting the release of muscle tensions [23], which are typical benefits of thermal aquatic exercise.

Although the effects of physical activity on psycho-social aspects of life associated with PD are still debated [11], the global score of the quality of life questionnaire for PD (PDQ-39) significantly improved after the thermal aquatic exercise, with the significance being driven mainly by the perceived mobility and activity of daily living dimensions that are strongly related to individuals’ motor status [41].

### Study Limitations

Although these results are promising, further studies on larger populations are surely worth performing. The greater effect of aquatic therapy versus land-based therapy on motor symptoms, assessed with the UPDRS scale and the Berg Balance Scale, have already been proved on a smaller cohort of six participants [18]. However, a randomized controlled trial aiming to evaluate the efficacy of the intervention described in this study compared to both land-based and conventional aquatic therapies is surely worth performing.

COVID-19 restrictions did not allow us to complete the whole evaluation protocol at the follow-up 2 (Figure 1), but results obtained from the six participants evaluated 1 month after the rehabilitation qualitatively show that their motor improvement did not completely wash out (Figure A1), as also reported in [68]: i.e., the 1 month follow-up scores for both Mini-BESTest and MDS-UPDRS part III indicate a worse motor condition than immediately after the rehabilitation but still improved with respect to the baseline.

Comparisons with other similar studies [19,20] are difficult due to the different chosen population. For instance, participants considered in [20] are much more compromised and a larger effect of aquatic exercise on motor status was then expected (i.e., TUG time at baseline in 20 equal to 13 s and minimum PDQ-39 global score equal to 60.3).

## 5. Conclusions

Thermal aquatic exercise may represent a promising rehabilitation tool to prevent the impact of motor symptoms on daily-life activities of people with Parkinson’s Disease, and to the risk of falls. The significance of comparison pre-post intervention on the PDQ-39 total score also foreshows good effects of thermal water intervention on quality of life and psychological well-being in people with PD.

## Figures and Tables

**Figure 1 healthcare-10-00368-f001:**
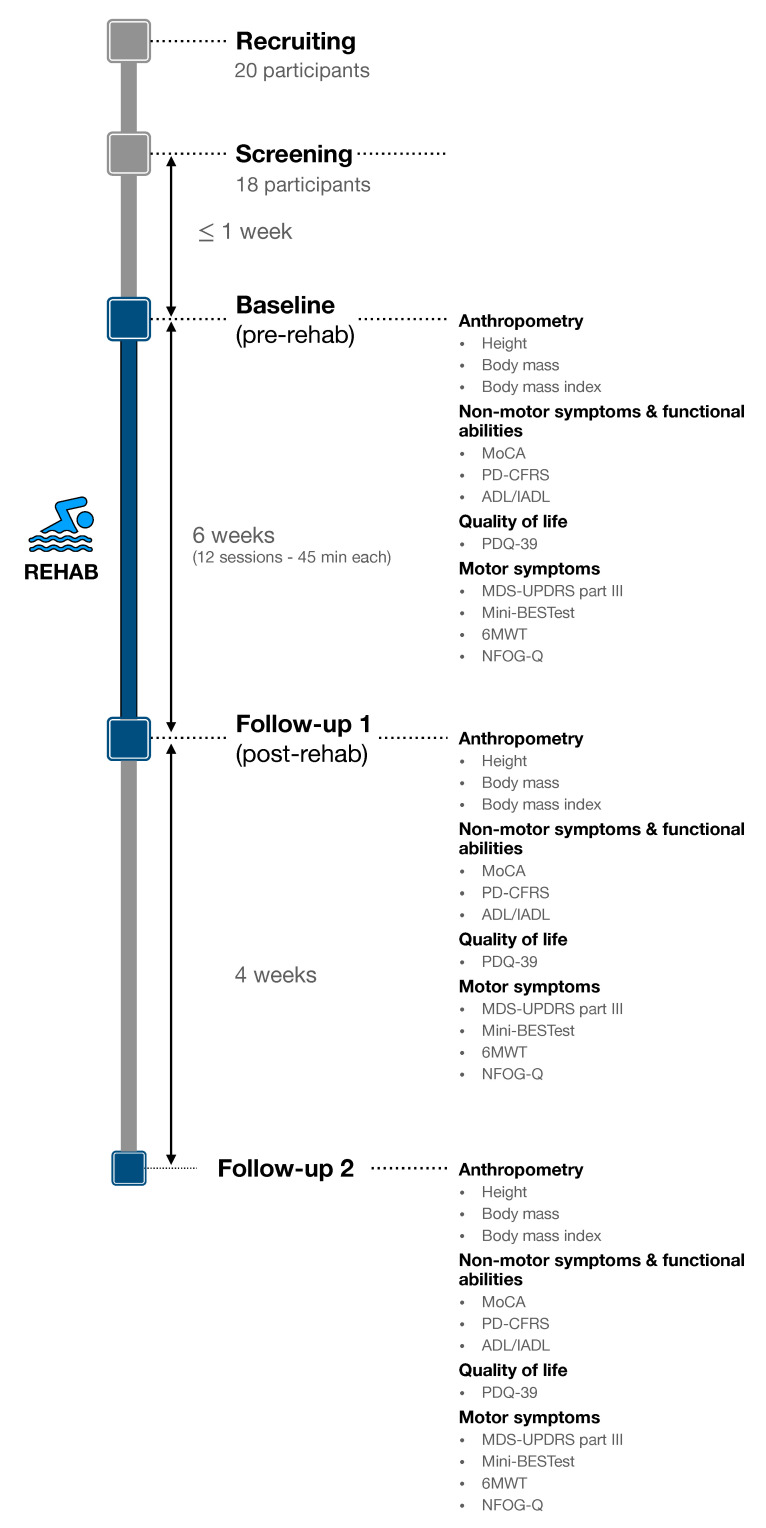
Evaluation protocol timeline.

**Figure 2 healthcare-10-00368-f002:**
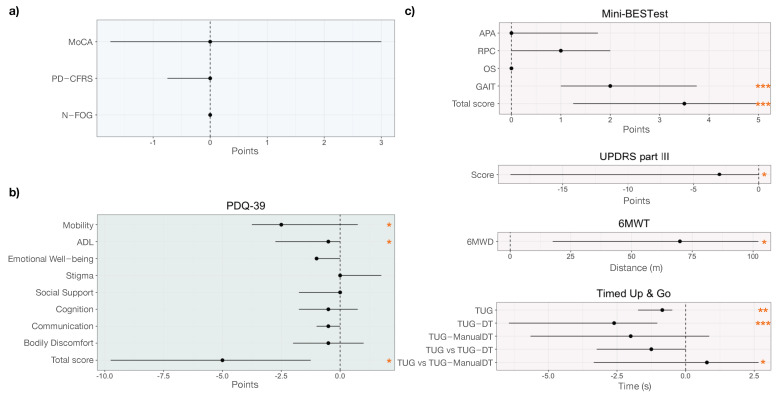
Forest plots highlighting the differences between baseline and follow-up visits for: (**a**) the Montreal Cognitive Assessment (MoCA), Parkinson’s Disease Cognitive Functional Rating Scale (PD-CFRS) and New Freezing of Gait questionnaire (N-FOG); (**b**) Parkinson’s Disease Questionnaire (PDQ-39) total score and subscores; and (**c**) Mini-BESTest (total score and subitems), the MDS-Unified Parkinson’s Disease Rating Scale (MDS-UPDRS) part III, the Six Minute Walk Distance (6MWD), and Timed Up and Go (TUG). Statistical differences are highlighted by the *.

**Figure 3 healthcare-10-00368-f003:**
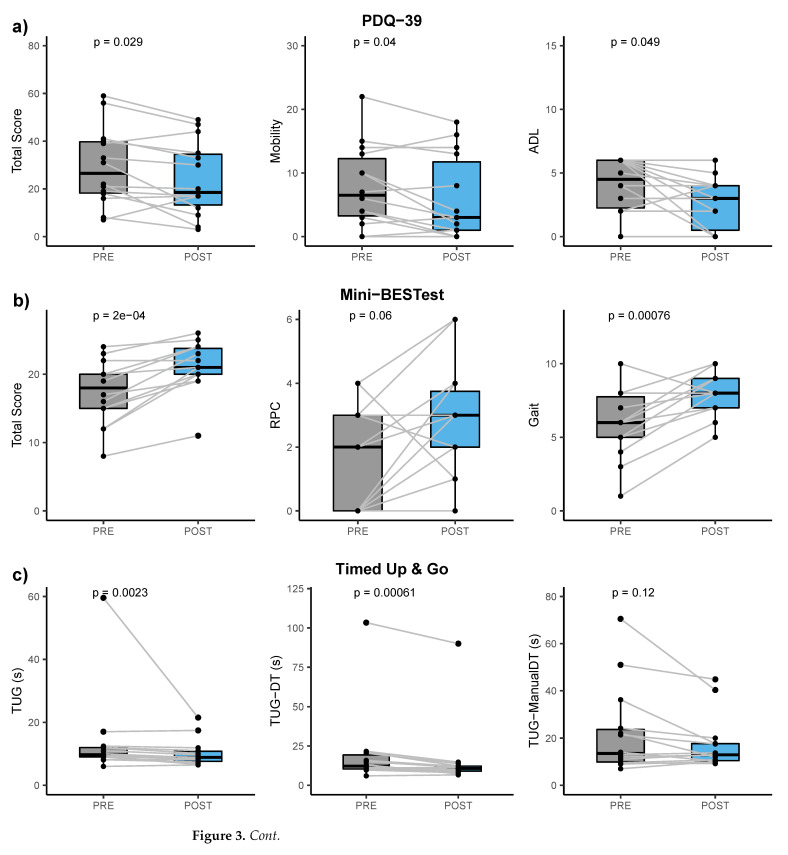
Box plots and spaghetti plots for: (**a**) the Parkinson’s Disease Questionnaire (PDQ-39) total score and Mobility and Activities of Daily Living (ADL) subscores; and the motor scales and tests that highlighted significant differences between baseline and follow-up visits, i.e., (**b**) the Mini-BESTest total score and the Reactive Postural Control (RPC) and Gait subscores; (**c**) the Timed Up and Go in single-task, cognitive dual-task, and motor dual-task duration; and (**d**) the six minute walk distance in meters (6MWD) and the Movement Disorders Society Unified Parkinson’s Disease Rating Scale—Part III score (MDS-UPDRS part III). Statistics is also reported on the box plots.

**Table 1 healthcare-10-00368-t001:** Thermal water exercise protocol.

Exercise Block	Description	Progression	Time	Repetition	Recovery
**Warm-up (10 min)**	Walking forward and backward	The 1st/2nd week it can be performed close to the wall Each week the difficulty will be increased (e.g., eyes closed condition)	5 min	-	1 min between each exercise
	Side stepping	N/A	2 min	-	
	Walking on the toes Walking on the heels	N/A	2 min	-	
**Main session (25 min)**	With participant sitting on a floating tubular and the physiotherapist standing behind him/her, participant performs a slow rocking movement side-to-side or front to back	N/A	2 min per direction	-	No rest
	Single-leg standing on each foot (starting from a gradually narrowing base of support)	The 1st/2nd week it can be performed close to the wall The last two weeks some turmoil can be added Progressively increase the unilateral stand time up to 60 s Add weights progressively during the last two weeks	1 min per foot	3 per exercise	No rest
	With both feet together and holding a flotation board, the trunk has to be turned in anterior-posterior and medial-lateral directions	N/A	15 min	-	No rest
	The therapist stands in front of the patient. She/he is asked to touch the therapist’s foot with her/his own foot Aquatic yoga—“the chair” position (knee flexion, hip flexion with arms stretched) and return to standing Walk with great steps by raising the knees and paddle forward Walk with 0.5 kg wristband Perform routes with rhythms: cha-cha-cha; back and forth with big steps (mark the rhythms out loud)	N/A The last two weeks “The tree position” should be performed N/A N/A			
**Cool-down (10 min)**	Stretch arms and leg with the help of the wall Passive flotation	N/A	5 min	-	No rest
	Supine position to relax muscles	N/A	5 min	-	

**Table 2 healthcare-10-00368-t002:** Social and demographic characteristic at the baseline. Data are given as median values with 1st and 3rd quartiles (25- and 75-percentile, respectively, Q1 and Q3), or as frequencies (N, %).

	PD Sample (*n* = 14) Median [Q1–Q3]/(Frequency)
Age (years)	70.5 [68.0–75.0]
Gender (male)	10 (71%)
Height (cm)	169.5 [165.0–170.8]
Body mass (kg)	72.5 [62.4–83.8]
BMI (kg/m${}^{2}$)	26.9 [23.0–29.3]
Education (years)	13.0 [10.0–15.3]
Age at disease onset (years)	62.0 [55.5–65.0]
Disease duration (years)	10.0 [6.0–12.5]
LEDD (mg/die)	770.0 [642.6–910.0]
DAED (mg/die)	131.0 [65.3–197.5]
H & Y	2 [2–3]
MMSE	29.0 [29.0–30.0]
Freezing of Gait	7 (50%)
Tremor	11 (78.6%)
Camptocormia	4 (28.6%)
Pisa syndrome	4 (28.6%)

**Table 3 healthcare-10-00368-t003:** Baseline and follow-up 1 evaluations. Data are given as median values with 1st and 3rd quartiles (25- and 75-percentile, respectively, Q1 and Q3). *p*-values calculated with the *t*-test are highlighted with a ${}^{†}$, the others are calculated with the Wilcoxon test. An * denotes statistical significance after Bonferroni correction (i.e., $p<p_{Bonferroni}=0.0027$).

	Baseline Median [Q1–Q3]	Follow-Up 1 Median [Q1–Q3]	Cohen’s $d_{z}$	*p*-Value
**MoCA** (corrected score)	23.85 [22.6–25.4]	24.98 [23.9–26.0]	0.159	0.2100
**ADL**	6.0 [6.0–6.0]	6.0 [6.0–6.0]	NA	NA
**IADL**	5.0 [5.0–5.8]	5.0 [5.0–5.8]	NA	NA
**PD-CFRS**	1.0 [0.0–1.0]	0.0 [0.0–1.0]	0.369	0.3150
**PDQ-39**	26.5 [18.3–39.8]	18.5 [13.3–34.5]	**0.668**	**0.0290 ${}^{†}$**
*PDQ-39 (Mobility)*	6.5 [3.3–12.3]	3.0 [1.0–11.8]	**0.627**	**0.0400**
*PDQ-39 (ADL)*	4.5 [2.3–6.0]	3.0 [0.5–4.0]	**0.266**	**0.0490**
**N-FOG**	3.0 [0.0–7.0]	2.0 [0.0–8.0]	0.225	0.8490
**MDS-UPDRS part III**	21.0 [12.0–33.8]	12.0 [11.0–16.0]	**0.753**	**0.0220**
**Mini-BESTest**	18.0 [15.0–20.0]	21 [20.0–23.8]	**1.362**	**0.0002 ${}^{†,}$***
*Mini-BEST (APA)*	4.0 [3.0–5.0]	4.5 [4.0–5.0]	0.426	0.1540
*Mini-BEST (RPC)*	2.0 [0.0–3.0]	3.0 [2.0–3.8]	0.590	0.0600
*Mini-BEST (OS)*	6.0 [6.0–6.0]	6.0 [6.0–6.0]	0.117	1.0000
*Mini-BEST (GAIT)*	6.0 [5.0–7.8]	8.0 [7.0–9.0]	**1.167**	**0.0008 ${}^{†,}$***
**TUG (s)**	9.7 [9.0–12.0]	8.9 [7.6–10.8]	**0.352**	**0.0023 ***
**TUG-DT (s)**	12.2 [10.3–19.3]	10.7 [9.0–12.2]	**0.973**	**0.0006 ***
**TUG-ManualDT (s)**	13.5 [9.9–23.7]	12.9 [10.4–17.7]	0.523	0.1200
**6MWD (m)**	531.5 [420.0–692.5]	600.0 [470.0–670.0]	**0.366**	**0.0250 ${}^{†}$**

## Data Availability

Anonymous data will be made available upon reasonable request to the Corresponding Author.

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
