# Peer review of "Effect of Intensive Rehabilitation Program in Thermal Water on a Group of People with Parkinson’s Disease: A Retrospective Longitudinal Study"

_healthcare, 2022, doi:10.3390/healthcare10020368_

Round 1

Reviewer 1 Report

This is a very interesting work in which the effects of thermal aquatic exercise have been tested on motor and non-motor symptoms of PD. The article is well-written, is comprenhensive and overall good quality. However, I do have some minor comments:

  1. ABSTRACT: Authors report that 20 iPD were recruited, but later in the text, they claim that from 21 patients only 14 finalized the therapy session. I would recommend to change the abstract section and say that "14 iPD patients were included".
  2. INTRODUCTION: The manuscripts starts (1st paragraph) by highlighting that the effect of physical therapy on non-motor symptoms is unknown. It seems that, given the position of this claim, the aim of the study was to assess the potential benefits on non-motor symptoms. Also, in the second paragraph, "aquatic-based" and "land-based" information is mixed. I think that the information that authors give in fine, but it should be reordered.
  3. In METHODS: the exclusion criteria are very well detailed, but the inclusion criteria not so much. How were patients diagnosed with PD? Which clinical criteria?
  4. The authors calcualted sample size prior to performing the study, and this was based on a non-parametric statistical test. However, looking at graphs it seems that some variables might follow a normal distribution, and therefore, parametric tests would be more appropriate in that case. Have the authors performed normality tests or plotted the data before performing the statistical analysis to chose the most appropriate test? (paired T-test instead of paired Wilcoxon test).
  5. Lines 74-75. The sentence is confusing and from Figure 1 cannot be intuitively understood. It states that two evaluations were carried out before and after session at the same day. Are the authors referring to pre and post evaluations (with 12 weeks interval)? Or are the authors claiming that evaluations at each timepoint were divided into two sessions? Or that all evaluations at each timepoint were carried out at the same day? It is not clear to me from the text. Please, clarify.
  6. Other minor issues:

Line 31: facilitates

Line 112-113: Difficult to understand. may be change to something like "... and the impact of cognitive impairment without dementia on daily functioning..."

Line 117: It seems that from the current writting style the authors claim that the questionnaire minimizes the motor impact, which I guess it is not what they meant. Please, change to something like "... but with a minimal impact of motor aspects of the disease" or similar.

Line 220: The authors report that "all patients from 5/14 received benefit from therapy, but n=3 remained at risk. Does this mean that only 2 out of 5 improved? If that's the case, please, rewrite the sentence.

Table 2. Usually, meadian and IQR are reported in the following format median[Q1 - Q3], instead of median (Q1;Q3). I would suggest changing the format. 

Reviewer 2 Report

This manuscript by Di Marco et al. is a nice follow-up to the previously published report by the same group (Masiero et al., 2019) and complements other current research showing the benefits of aquatic exercise in the treatment of persons afflicted with Parkinson’s Disease. The main focus of this retrospective and longitudinal study is on the effects of rigorous thermal aquatic exercises on motor and non-motor PD symptoms. The manuscript is easy-to-read, the appropriate statistical tests were performed and the findings are of interest to other investigators in the field of neurodegenerative diseases. This reviewer has only a few minor comments that should be addressed prior to publication:

The authors may want to consider including more references to their previous, closely related study published in 2019.

Pg. 2, line 38. I believe [21] is incorrectly cited here. Please provide an appropriate reference to that explains the benefits of thermal aquatic exercise.

Pg. 2, lines 56-62. A brief explanation/rationale for the chosen exclusion criteria should be included.

Pg. 4, line 98. Instead of saying “…doses were calculated according to [33]” consider “…doses were calculated according to Tomlinson et al. (2010) [33]"

Pg. 15 (Figure 2). In (c), there appears to be an outlier in the group. Can the authors provide reasons for the outlier?

Grammatical criticisms:

Pg. 1, line 26. “land based” should be “land-based”

Pg. 2, line 31. “facilitate” should be “facilitates”

Pg. 2, line 33. It’s unclear what the authors are trying to say here…perhaps consider changing “reach of” to “rich in”

Pg. 2, line 44 (and Pg. 7, line 225). Add hyphenation for consistency. “non-motor”

Pg. 4, line 103. “post rehabilitation” should be ‘post-rehabilitation”

Pg. 7, line 229. “…did not have impact” should be “…did not have an impact”

Pg. 7, line 240. Remove “being”

Pg. 8, line 267. “…allow us completing” should be “…allow us to complete”

Reviewer 3 Report

Di Marco et al. report a small but interesting experimental uncontrolled trial which assessed the effect of thermal aquatic exercise on motor function and quality of life in 14 PD patients. The authors show that an intensive rehabilitation program in thermal water (12 sessions in 6 weeks) improved balance, motor function and quality of life.

The study is clear and well conducted. However, I have some major and minor remarks:

Major remarks:

- Throughout the manuscript there is an inaccurate overlap between the concepts of quality of life and non-motor symptoms in PD. Although psycho-social aspects and low quality of life can be direct consequences of non-motor symptoms in PD they are not considered non-motor symptoms themselves. Please address this issue.

- I have some concerns about the statistical analysis (not my area of expertise). Considered that this study evaluates multiple endpoints, shouldn't statistical tests be corrected for multiple testing (adjusted p-value)?

Minor remarks:

Abstract:

1 - I suggest modifying the starting sentence as follows: “The main objectives of the study are to test the effect of thermal aquatic exercise on motor symptoms and quality of life”.

2 – The number of participants (n=14) that were included and completed the whole evaluation and rehabilitation protocol should be also stated here.

10 - I would modify this sentence as follows: “Balance of all the participants significantly improved (Mini-BESTest: p < 0.01)”.

Introduction:

22 – Please correct the improper overlap between the concepts of quality of life and non-motor symptoms.

46 – Which non-motor symptoms do the authors refer to in this sentence? Please clarify.

Methods:

151 – The main outcomes of the study should be stated clearly here.

Results:

173 – Were the study participants 20, as stated in the abstract, or 18 as stated here?

Discussion:

224 – This study is aimed at investigating the effect of thermal exercise on motor symptoms, functional abilities, and quality of life more than non-motor symptoms (only cognitive performance was quantitatively evaluated and was preserved in all participants).

235 – Why the authors state that the thermal aquatic exercise seemed to have a positive impact on functional abilities? The functional status as assessed by ADL and IADL was similar between baseline and after rehabilitation. Please clarify.

Conclusions:

281 – I suggest substituting “non-motor symptoms associated with PD” with “quality of life in PD patients”.

Round 2

Reviewer 3 Report

The manuscript improved in the revised version. I have no major remarks. 

Author Response

Thank you very much for your feedback.